# Chemically Reduced Graphene Oxide-Reinforced Poly(Lactic Acid)/Poly(Ethylene Glycol) Nanocomposites: Preparation, Characterization, and Applications in Electromagnetic Interference Shielding

**DOI:** 10.3390/polym11040661

**Published:** 2019-04-11

**Authors:** Ahmad Fahad Ahmad, Sidek Ab Aziz, Zulkifly Abbas, Suzan Jabbar Obaiys, Khamirul Amin Matori, Mohd Hafiz Mohd Zaid, Haider K. Raad, Umar Sa’ad Aliyu

**Affiliations:** 1Department of Physics, Faculty of Science, Universiti Putra Malaysia, Serdang 43400, Malaysia; za@upm.edu.my (Z.A.); khamirul@upm.edu.my (K.A.M.); mhmzaid@upm.edu.my (M.H.M.Z.); 2School of Mathematical & Computer Sciences, Heriot-Watt University Malaysia, Putrajaya 62200, Malaysia; s.obaiys@hw.ac.uk; 3Engineering Physics Program, Xavier University, Cincinnati, OH 45207, USA; raadh@xavier.edu; 4Department of Physics, Federal University Lafia, Lafia 0146, Nigeria; usaltilde@yahoo.com

**Keywords:** PLA/PEG blend, reinforcement, polymers, nanocomposites, adhesion, dielectric

## Abstract

In this study, a nanocomposite of reduced graphene oxide (RGO) nanofiller-reinforcement poly(lactic acid) (PLA)/poly(ethylene glycol) (PEG) matrix was prepared via the melt blending method. The flexibility of PLA was improved by blending the polymer with a PEG plasticizer as a second polymer. To enhance the electromagnetic interference shielding properties of the nanocomposite, different RGO wt % were combined with the PLA/PEG blend. Using Fourier-transform infrared (FT-IR) spectroscopy, field emission scanning electron microscopy (FE-SEM) and X-ray diffraction, the structural, microstructure, and morphological properties of the polymer and the RGO/PLA/PEG nanocomposites were examined. These studies showed that the RGO addition did not considerably affect the crystallinity of the resulting nanomaterials. Thermal analysis (TGA) reveals that the addition of RGO highly improved the thermal stability of PLA/PEG nanocomposites. The dielectric properties and electromagnetic interference shielding effectiveness of the synthesized nanocomposites were calculated and showed a higher SE total value than the target value (20 dB). On the other hand, the results showed an increased power loss by increasing the frequency and conversely decreased with an increased percentage of filler.

## 1. Introduction

Electromagnetic interference (EMI) is one of the most undesirable byproducts of the rapid growth of telecommunication devices and high-frequency electronic systems. Any device that utilizes, distributes, processes, or transmits any form of electrical energy is likely to interfere with nearby equipment or systems’ operation and emit electromagnetic signals [1]. Negative effects on the health of human being might also be a result of such phenomenon. There has been a great deal of effort invested in the reduction of electromagnetic pollution using EMI shielding materials [2], where signals are attenuated by the shielding materials through absorption and/or reflection of the radiation power [3]. Magnetic materials and metallic structures have been used traditionally for EMI shielding because of their high effectiveness in shielding as well as desirable mechanical properties. However, several setbacks which include difficult processability, heavy weight, and exposure to corrosion have been identified with the materials involved in such shielding. Also, the metals applied as absorbers of electromagnetic radiations are limited by their high conductivity, since they have a shallow skin depth and thus most of the radiation power on the surface gets reflected [4]. Thus, researchers’ interests have been escalated to develop practical and effective materials for EMI shielding to overcome the shortcomings of the conventional metal-based shields [5]. Electrically conductive polymers and polymer composites have attracted significant attention for applications in EMI shielding. These materials are fabricated to be lightweight, resistant to corrosion, easily processable, and flexible, which addresses the drawbacks of metals [6,7].

The majority of the polymers used commercially, such as polypropylene, polystyrene, and polyethylene, are non-biodegradable. Due to both economic and environmental concerns associated with petroleum production expenses and waste disposal, there has been ever-increasing interest in both biosourced and biodegradable polymers [8]. Poly(lactic acid) (PLA) is linear, aliphatic thermoplastic polyester. It is derived from renewable sources, such as a corn starch, [9] and is completely biodegradable. On the list of bioplastics, with its commercial availability, thermal plasticity, and reasonable price, PLA is placed on the top of the list [10]. PLA is also a potential alternative to polypropylene and polyethylene (traditional non-biodegradable polymers), since it was more cost effective and recyclable. However, its wider applications are limited by some of its disadvantages, which include slow crystallization rate, relatively poor mechanical properties and gas barrier, and low thermal stability [11]. Therefore, the preparation of nanocomposites and polymer blending has to often undergo a modification of the PLA physical properties for the extension of their practical applications [12]. Several nanoreinforcement fillers, such as layered double hydroxide [13], carbon nanotubes [14], and layered silicate clay [15], have been developed and extensively studied in various polymer matrices. These attempts include PLA modification with plasticizers, PLA blending with other polymers, or PLA blending with other inorganic nanofillers [16].

Macrogols is another form of polyethylene glycols (PEG), which can either be solid or liquid polymers with H(OCH_2_CH_2_)*n*·OH as general formula. As PLA’s plasticizing agent, PEG has shown a great potential as it provides a considerable increase in chain elongation at break [17]. Through glass transition temperature (*T*_g_) reduction, PEG acts as a plasticizer to the PLA. The addition of PEG-200 in the system has led to an increase in the tensile properties and elongation at break (>7000%). However, this was also accompanied by a reduction in tensile modulus and tensile strength. However, the mixture (PLA/PEG) has shown no stability and lost its beneficial properties over time due to the separation of phase at ambient temperatures, and therefore leading to the PLA-rich and PEG-rich phases’ formation [18].

In terms of electromagnetic properties, very few articles have reported the EMI shielding properties of PLA nanocomposites. Recently, Frackowiak et al. [19] reported the EMI shielding performance of foamed PLA nanocomposites containing carbon black and carbon fibre in the frequency range of 100 Hz to 1 MHz. Graphene has attracted tremendous attention since it was first experimentally prepared in 2004 [20] as a novel carbon nanoparticle, alongside other carbon-based particles such as, carbon black [21], carbon fibre [22], and carbon nanotubes [23]. The unparalleled combination of properties, including large surface area, high electrical conductivity, high mechanical properties, and low cost, has facilitated graphene to become one of the potential choices to prepare multifunctional composites; moreover, graphene has been used in several studies for preparation of EMI shielding materials [24] with the possibility for reinforcing polymers and improving new materials with multifunctional properties. For polymer matrices’ reinforcement, graphene has widely been used due to its high thermal conductivity, large specific surface area, good thermal stability, superior mechanical properties, etc. [25]. Four effective methods of graphene preparation include [26]: epitaxial growth [27], micromechanical exfoliation of graphite [28], chemical vapour deposition [29], and chemical exfoliation [30]. For large-quantity production of graphene, chemical exfoliation is the most feasible method. In this method, chemically reduced graphene oxide (RGO) (graphene), was prepared through oxidizing graphite in the presence of oxidants and strong acids, followed by graphene oxide (GO) reduction using one of the reduction chemical methods [31]. On its surface, the residual oxygen functional groups were retained by the resulting RGO, which will greatly influence its properties and that of its nanocomposites [32].

The electromagnetic properties and EMI shielding effectiveness of nanocomposites depend on the reflection from the material’s surface, absorption of the EM energy, and propagation paths of the EM wave, which are determined by the nature, shape, size, and microstructure of the fillers [33]. There are some studies that have investigated and contributed to the knowledge of the effect of the particle size of the graphene oxide on the electromagnetic properties, where the properties can be tuned by controlling size of the graphene particles. This phenomenon may be interpreted as follows: the interfacial polarization is dominant in the heterogeneous structures, which contributes to the various electromagnetic properties [34]. Also, defects of graphene oxide act as dipolar polarization centers, leading to various dielectric relaxations in different frequency ranges. With decreasing nanoparticle size, larger interface area and more defects exist in the nanoparticles, which enhance the electromagnetic properties of the nanoparticles [35].

In this work, the RGO/PLA/PEG nanocomposites’ preparation was carried out by using the technique called melt blending using a Brabender internal mixer. The properties of the RGO/PLA/PEG nanocomposites, including the crystallization behaviour, the functional groups of the nanocomposites, the structure of the composite, and the thermal behaviour of the composites, are thoroughly investigated. Furthermore, the effectiveness of the EMI shielding properties of nanocomposites over the X band (8–12 GHz) is also investigated. It was found that the applied frequency and the filler concentrations have affected all the essential properties of the RGO/PLA/PEG nanocomposites.

## 2. Experimental Details

### 2.1. Materials

The main material utilized in this study is the PLA polymer matrix pellets with a density of 1.24 g/cm^3^ (Grade 4060D) from Nature Work LLC (Minnetonka, MN, USA). Low molecular weight polyethylene glycol (PEG) (*M*_n_ = 200 g/mol) was acquired from Sigma-Aldrich (St. Louis, MO, USA). Reduced Graphite Oxide powder was synthesized via a chemical method in the lab. NH_3_ was supplied by Sigma Aldrich (Sarasota, FL, USA). The chemical structures of the materials under test are illustrated in Figure 1.

### 2.2. Preparation of Reduced Graphene Oxide (RGO)

Reduced graphene oxide (RGO) powder was synthesized by a chemical method with an average size of 60 nm. The chemical method of manufacturing and preparing the RGO powder involved two major steps. The first step is the synthesis of Graphite Oxide (GO) using the Staudenmaier Method [35]. The second step performs the reduction process of GO to RGO. About 400 mg of the obtained GO was placed in a cellulose extraction thimble (30 by 100 mm) and was then placed in the Soxhlet extraction unit. Approximately 150 mL of 30% Ammonia solution (NH_3_) was used as a reducing agent. The heating temperature was set at 90 °C, and the investigated exposure period was 5 h when the GO powder had direct contact with the ammonia vapor as well as the condensed liquid. The procedure of manufacturing and preparation is shown in Figure 2.

### 2.3. Preparation of RGO/PLA/PEG Nanocomposites

To remove the moisture content of PLA during blending, both PLA pellets and RGO powder were dried at 80 °C for 12 h in a vacuum oven before processing [36]. The RGO/PLA/PEG nanocomposites’ preparation was carried out by using the technique called melt blending for 10 min using Brabender internal mixer (GmbH & Co. KG, Duisburg, Germany) at 170 °C with 60 rpm rotor speed [37]. The PLA to PEG weight ratio was kept constant at 90/10 wt/wt. The RGO was added as a filler at a different percentage with the PLA/PEG blend as presented in Table 1. The dispersion of RGO nanofiller within the PLA/PEG polymer leads to network formation as a result of more physical contacts between the particles of filler and matrix inside the RGO/PLA/PEG nanocomposites. The structure of RGO/PLA/PEG nanocomposites is schematically illustrated in Figure 3.

The molding of the obtained composite into sheets of thickness 1 mm was carried out by hot pressing for 10 min at 170 °C with a pressure of 110 k/bar, followed by room temperature cooling. The prepared plates were used for further characterization. As for the electromagnetic properties of nanocomposites, the dough was molded into rectangular-shaped specimens with a thickness of 6 mm by hot pressing. The preparation temperature was 170 °C and the pressure force was kept at 110 k/bar for 10 min, after which the specimens were allowed to cool down for 10 min at a force of 110 k/bar to a temperature of 50 °C.

The rectangular mold specimen was placed between the two rectangular waveguides to minimize the air gap between waveguide walls and the border of the specimen. Figure 4a,b illustrates the process used in fabricating the substrates composites and the specimens where the mixture was poured into rectangular aluminum molds of 6 mm thickness, in preparation for the electromagnetic measurements of composites.

### 2.4. Characterisation

#### 2.4.1. Morphological Characterization 

FE-SEM images provide in-depth information which reveals the characteristic 3-D manifestation necessary for understanding the morphology of the cross-sectional surface of a sample. Accordingly, the surface morphology of RGO/PLA/PEG nanocomposites and the dispersion of RGO nanoparticles in the PLA/PEG matrix were studied using FE-SEM (FEI Quanta 200 SEM, Yuseong, Daejeon, Korea) at a fixed voltage of 10 kV. The specimens were dried for 45 min before being coated with gold particles using a SEM coating unit (Baltic SC030 sputter coater, Yuseong, Daejeon, Korea).

#### 2.4.2. X-Ray Diffraction

The measurement of X-ray diffraction was performed by using an X-ray diffractometer (XRD, XD-3, Cu Ka radiation) under ambient conditions with a Lynx Eye detector using a Bruker diffractometer ((Yuseong, Daejeon, Korea) over 2θ range of 5°–80°. The used X-ray beam was Cu-Kα radiation (nickel-filtered) with 1.54 Å as wavelength, operated under a voltage of 40 kV and 35 mA current. A fine powdered form of RGO and cut samples of compression molded specimens of PLA and RGO/PLA/PEG nanocomposites were used. The RGO nanoparticles’ average crystallite size (≈60 nm) was calculated using the Scherrer Equation:(1)D=kλ/β.cosθ
where the average crystallite size is given as D (in nm), the shape factor (normally 0.9 for cubic) is k, the full-width half maximum intensity measured diffraction line broadening is β (FWHM data converted to radians), Bragg’s diffraction angle is θ, and the X-ray’s wavelength is λ.

#### 2.4.3. Thermal Stability (TGA and DTG) Properties

Thermogravimetric analysis (TGA) analysis is a technique in which a measurement of the mass of a substance is carried out depending on a temperature or time function while the material undergoes a controlled temperature program. TGA is also useful for compositional analysis of multicomponent materials and used to examine the kinetics of the physiochemical processes occurring in the sample, whereas the Derivative thermogravimetric (DTG) thermograms were utilized for composites’ weight-loss study. The investigation of the RGO/PLA nanocomposites’ thermal stability was performed using a TGA (1600LF, Shanghai Mettler Toledo Co. Ltd., Shanghai, China). The RGO/PLA specimens, weighing 5–10 mg each, were heated from 50 °C to 600 °C at a heating rate of 10 °C/min^−1^ under a nitrogen atmosphere with a flow rate of 20 mL/min. The instrument was computer controlled while Pyris software was used for calculations. The instrument was calibrated using the Curie temperatures of five different metal standards.

#### 2.4.4. Fourier Transform Infrared (FT-IR) Analysis

A Perkin-Elmer FT-IR 1650 spectrophotometer (Waltham, MA, USA) was used for the FT-IR characterization, identification of chemical bonds in a molecule, and determination of the functional groups in the net and RGO/PLA/PEG nanocomposites at different percentage of RGO. The FT-IR spectroscopy involves the collection of information on IR absorption and analyzing them in spectrum form by correlating the frequencies of IR radiation absorption (“peaks” or “signals”) directly to the bonds present in the compound under investigation. Using a KBr disk method, the FT-IR spectra tests were carried out in the 400 to 4000 cm^−1^ wavenumber range, at room temperature.

#### 2.4.5. Electromagnetic Interference (EMI) Properties

##### Electrical Properties

The dielectric properties and Scattering (S-parameters) reflection (S11) and transmission (S21) coefficients were measured using a transmission line technique (rectangular waveguide) by commercial measurement software on the Agilent N5230A PNA-L network analyzer system which includes the Agilent 85071E, 85701B software package, respectively [38] (Agilent Tech, 2010, Keysight Technologies, Santa Rosa, CA, USA), in the 8–12 GHz frequency range. The rectangular waveguide connected to the vector network analyzer (VNA) has been calibrated by full two-port, Thru-Reflect-Line (TRL) [39].

##### EMI Shielding Effectiveness (EMI SE)

The electromagnetic interference shielding effectiveness (EMI SE) is defined as the ability of a shield material to reduce the electromagnetic field. It can also be defined by the ratio of incoming to outgoing power. The total EMI SE of a material is specified to take place by the three mechanisms namely: absorption shielding effectiveness (${\mathrm{SE}}_{A}$), reflection shielding effectiveness (${\mathrm{SE}}_{R}$), and multiple internal reflections (${\mathrm{SE}}_{M}$). A schematic mechanism of EMI SE is depicted in Figure 5. The scattering parameters (S_11_, S_12_, S_21_, and S_22_) were obtained as shielding components for the conductive nanocomposites. The shielding effectiveness for the nanocomposites has been calculated by using Equation (2) [40]. The absorption mechanism is associated with the dielectric/magnetic polarization or energy dissipation [41]. The reflection, which is considered as the primary mechanism of shielding, involves the interactions between the electromagnetic fields and the charge carriers, such as electrons and holes. It can also be related with the impedance mismatch between the air and an absorber. Internal multiple reflections are related to the reflection between the opposite faces of a material. Therefore, in order to barricade the electromagnetic wave, the shielding material should either absorb or reflect the wave. In general, EMI SE is expressed in decibel (dB) units.

The EMI SE of a material can be expressed using the following equation [40]:
(2)$${\mathrm{EMI}\;\mathrm{SE}}_{\mathrm{total}}\left(\mathrm{dB}\right)=10*\log\left(\frac{P_{\mathrm{in}}}{P_{\mathrm{tr}}}\right)={\mathrm{SE}}_{A}+{\mathrm{SE}}_{R}+{\mathrm{SE}}_{M}$$
where P_in_ and P_tr_ are power of incident and transmitted EM waves, respectively. In the case of a thick shield, the SE_M_ can be neglected due to high absorption loss. When, for the second time, the wave reaches the second boundary, its amplitude becomes negligible, due to the fact that it has passed through the shield’s thickness three times by then. The EMI SE equation can be written as [42]:(3)$${\mathrm{SE}}_{\mathrm{total}}={\mathrm{SE}}_{A}+{\mathrm{SE}}_{R}$$

Therefore, experimental absorption and reflection losses can be expressed as [43]:
(4)$${\mathrm{SE}}_{R}=10*\log\;(1/(1-S_{11}^{2}))$$
(5)$${\mathrm{SE}}_{A}=10*\log\;((1-S_{11}^{2})/S_{21}^{2})$$
where the S_11_ and S_21_ scattering parameters are coefficients of reflection and transmission, respectively.

## 3. Results and Discussion

### 3.1. Field Emission-Scanning Electron Microscopy Results

The results of FE-SEM were used for the study of the fractured tensile specimens’ surface morphology and to visualize qualitatively the RGOs’ dispersion state in the matrix of the PLA/PEG mixture. The surface micrographs of RGO/PLA/PEG nanocomposites, PLA/PEG mixture, and the neat PLA are shown in Figure 6a–f. Figure 6a shows a typical tensile fracture surface of PLA. Thermoplastic organizing of the PLA after melting is accredited to real melting processes due to its brittle behavior at room temperature, as shown in the Figure [35]. Illustrated in Figure 6b, the PLA/PEG mixture is subjected to the process of deformation and a few long threads of deformed material are discernible on the fracture surface of the sample. Figure 6c clearly shows the layered, porous, wrinkled silk-like morphology; the layered structure was formed by RGO particles and continually cross-linked in a flaky textured form, as described by [44].

To understand the dispersion of RGO in the PLA/PEG mixture, the wrinkled morphology of RGO is assumed to be necessary in the mechanical interlocking with the polymer matrix, which helped to build a strong interfacial interactions and subsequently efficient stress transfer across the interface [45]. The RGO particles are located on the surface of PLA/PEG matrix and trapped between the matrix like a sandwich in some areas, indicating an electrostatic attraction between RGO powder and matrix, which may contribute to the microwave absorption [46].

Good composite uniformity indicates a good dispersion degree of RGO and thus results in good thermal stability and tensile properties. On the other hand, after increasing the percentage of RGO (2.4% and 4%) in the PLA/PEG mixture, the dispersion of RGO powder in the matrix in Figure 6e,f showed obvious differences. Figure 6e showed that the RGO powder was dispersed in the PLA/PEG mixture as merged particles of a large size. Figure 6f showed that the RGO powder was dispersed in the PLA/PEG mixture in the form of agglomerates. The outlines of the PLA/PEG mixture and RGO nanoparticles are clearly observable. In addition, it can be seen that the small RGO particles have a propensity to aggregate due to their inherent properties. However, it is considered that suitable RGO nanoparticle content could cause uniform distribution at the matrix and suppress the aggregation.

### 3.2. Fourier Transform Infrared (FT-IR) Analysis

Figure 6 presents the FT-IR spectra for pure material, PLA/PEG mixture, and RGO/PLA/PEG nanocomposites. The PLA spectrum shows four main regions corresponding to (1100–1000) cm^−1^, (1500–1400) cm^−1^, (1750–1745) cm^−1^, and (3000–2850) cm^−1^, respectively. The PEG spectrum shows a clear peak at the absorption bands corresponding to 3441 cm^−1^, 2878 cm^−1^, 1464, 1343, and 1279 cm^−1^, respectively [47]. Figure 7 shows the RGO spectrum, with four main regions of absorption bands of the RGO powder, which correspond to 1039.46 cm^−1^, 1388.03 cm^−1^, 1494.24 cm^−1^, 1590.97 cm^−1^, 3223.46 cm^−1^, and 3389.86 cm^−1^, respectively [35]. The distinguishing peaks responsible for –C=O stretching, –CH stretching, –C–O stretching, as well as C–H bending were clearly observed for all RGO/PLA/PEG nanocomposites over the spectra and no new peaks were formed. Therefore, a conclusion can be made that no chemical interaction took place in the polymer matrices with RGO addition. This is expected since RGO does not have any functional groups with which to form a strong interface with a polymer matrix. Thus, any change in the nanocomposites’ properties is the effect of the physical interaction only between the PLA/PEG mixture and the RGO powder. It was also clear from the results that no considerable change was observed in the peak positions of the RGO/PLA/PGE nanocomposites compared to pristine PLA/PGE mixture.

### 3.3. X-Ray Diffraction

Figure 8 illustrates the X-ray diffraction pattern of the PLA/PEG mixture, RGO powder, and RGO/PLA/PEG nanocomposites at different percentages of RGO filler. For the PLA/PEG mixture, a broad amorphous peak from PLA was observed which is the typical peak for any given amorphous structure around 17.48°, this is in agreement with [48]. Figure 7 confirms the X-ray diffraction pattern of the RGO powder, showing a good crystallinity. The curve indicates a series of diffraction peaks at 2θ = 17.85°, 38.57°, 42.23°, 44.87°, and 73.02°, which correspond to the (d002), (d100), (d101), (d102), and (d004) planes, respectively [35]. Normally, the disappearance of the characteristic peak for RGO powder in the RGO/PLA/PEG nanocomposites can be correlated to fully exfoliated RGO powder in the polymer matrix. This indicates that the accumulated layers of RGO have exceeded high shearing during melt-mixing. The RGO/PLA/PEG nanocomposites at all percentages showed almost the same absorption peaks as pristine PLA. This means that there is no new bond created or strong chemical interaction occurring between the matrix and the nanocomposites.

### 3.4. Thermogravimetric Analysis (TGA)

During its service life, an EMI shielding material may be subject to high-temperature conditions. In this study, the thermal stability of the composites was analyzed. A feasible EMI shield should have suitable thermal stability, so that it may perform well at elevated temperatures [49]. Reduced graphene oxide has been applied widely as a filler to improve the thermal stability of the polymer matrix. In this research, thermogravimetric analysis (TGA) and the curves of the derivative (DTG) thermogravimetric are used to gain information about the extent and nature of the material degradation under test and RGO/PLA/PEG nanocomposites. The detailed experimental results are shown in Figure 9 and Table 2. The TGA results illustrated the PEG/PLA matrix showed a lower thermal stability compared to pure PLA. The reduction in the thermal stability of PLA is mainly a result of PEG’s presence as a plasticizer. PEG causes a reduction in thermal stability by its action to intersperse itself around polymers and by breaking the interaction between the PLA and PEG polymers, which is predicted by the gel theory of plasticization and lubricity theory [50].

The thermal degradation temperature of pure RGO powder, pure PLA, and RGO/PLA/PEG nanocomposites is between 227 °C and 400 °C. The thermal decomposition curves of RGO was at 227 °C, while PLA was at 356 °C. The thermal decomposition curves of RGO/PLA/PEG nanocomposites at different percentage of filler shifted towards higher temperatures, compared to the pure PLA/PEG matrix with increasing RGO content. The 50% weight loss temperatures (T_50%_ °C) of pure PLA was 365 °C, while at 0.8%, 2.4%, and 4% RGO content, they were 330 °C, 340 °C, and 350 °C, respectively.

Figure 9 and Table 2 show that the T_d-max_, which represents maximum decomposition temperature, also shifted towards higher temperatures with increasing RGO content, and the Td-max = 386.167 °C of the composites was increased with 4.0 wt% RGO filler compared to pure PLA polymer. This suggests that RGO nanoparticles can control the thermal stability of the RGO/PLA/PEG nanocomposites as well as T_50%_ and T_d-max_. At the beginning, the degradation temperature of the composites can be ascribed to the early decomposition of RGO in the matrix. At 50% weight loss, the thermal decomposition temperature of the composites was found to have improved. Generally, RGO incorporation into the PLA/PEG mixture enhanced the thermal stability by acting as a superior insulator and mass transport barrier to the volatile products generated during decomposition. Many researchers have also demonstrated that the incorporation of graphene or its chemical derivatives could enhance the thermal stability of PLA at extremely low loading contents [51]. It has to be noted that the PLA/PEG blends and RGO/PLA/PEG nanocomposites showed lower degradation temperatures compared to the neat PLA due to the thermal decomposition of the polymer matrix [52].

Table 2 shows weight loss (%) at T_d-max_ temperatures. It was observed that the weight loss (%) of pure PLA, PLA/PEG, RGO, and its nanocomposite (0.8% RGO, 2.4% RGO, and 4% RGO) at 400 °C were 97%, 95%, 27%, 92%, 90%, and 87% respectively. The delayed degradation of the PLA/PEG chain with increasing concentration of RGO led to the enhanced thermal stability of the nanocomposites. It was observed that the weight loss % of RGO at T_d-max_ was 27% because the thermal decomposition of RGO starts slightly below 400 °C, and maximum decomposition occurs above 600 °C, leading to the enhancement of thermal stability.

Figure 10 shows the derivative thermograms (DTG) of pure PLA, PLA/PEG mixture, and RGO/PLA/PEG nanocomposites at different percentages of filler, while the RGO performs as a heat barrier, which enhances the overall thermal stability of the nanocomposites. It also assists in the formation of ash after thermal decomposition. The RGO shifts the decomposition to a higher temperature at early stages of thermal decomposition. The improvement in thermal stability can be attributed to the “tortuous path” effect of RGO, which retards the escape of volatile degradation products. The presence of high RGO loading or well-dispersed RGO in the polymer matrix will force the degradation products to go through the more tortuous path and hence enhanced the thermal stability. Similar results have been reported for other graphene-based nanocomposites [53].

### 3.5. The Dielectric Properties of the Composites

The permittivity (ε) of a material determines the material’s response to the electric field component of the electromagnetic wave. Permittivity is determined by the complex term ε = ε′ − jε″. Insulating polymers have low permittivity due to the small degree of polarization of the macromolecules. Addition of conductive fillers to polymers can lead to considerable improvement to the low permittivity of the matrix [54] since the polarization of filler and polarization of filler/polymer interface (interfacial polarization) can contribute significantly to the overall polarization of the composite. When a current flow across the interface of two materials, it can cause accumulation of charges at the interface due to the difference in the materials’ relaxation times and consequently increases its permittivity [55].

Figure 11a,b shows the real and imaginary part of permittivity of the RGO/PLA/PEG nanocomposites versus RGO loading. It is evident that the permittivity is very sensitive to RGO loading. Both the real and imaginary parts of the permittivity are found to have increased with increasing RGO concentration. The real part increases reasonably from 2.75 to 3.79 as the RGO loading is raised from 0 to 4 wt% and the imaginary part increases from 0.093 to 0.50 for the same increment in filler content. This significant improvement in both dielectric constant and loss factor is the result of an increase in conductivity and dipole moment of RGO/PLA/PEG nanocomposites due to addition of conductive nanoparticles [56].

The permittivity of the composites is related to the absorption of an electromagnetic wave along with the dielectric constant and the thickness of materials. It is anticipated that the conductive network formed by the RGO interacts with the entering power signal and assists the movement of an electron within the composites, thus being responsible for the absorption of incident power. Also, the interaction between the RGO and PLA/PEG matrix contributes to the movement of electrons in the composites. It is expected that the conducting RGO renders multiple interfaces that increases the reflection and greatly barricades the electromagnetic wave inside the composites. The fine dispersed RGO particles facilitate easy movement of free electrons inside the insulating polymer matrix, even at low filler loading.

### 3.6. EMI Shielding Effectiveness

The EMI SE of RGO/PLA/PEG nanocomposites was determined in the frequency range of 8–12 GHz which is illustrated in Figure 12a–d. The EMI SE values of the RGO/PLA/PEG nanocomposites were calculated using Equations (2)–(5), respectively. The PLA/PEG matrix is transparent to the electromagnetic radiation and does not show any EMI shielding efficiency due to the very low permittivity of pure PLA. The SE_total_ of the nanocomposites increases upon increasing RGO loading due to the enhancement of permittivity (ε′ and ε″). With the increase in filler loading, the EMI shielding efficiency of RGO/PLA/PEG nanocomposites increases. The EMI SE value of 22.5 dB was recorded at 0.8 wt % RGO loading. The high value of EMI SE obtained at low loading of functionalized RGO can be attributed to the fine distribution and dispersion of RGO in the PLA/PEG matrix, thereby forming an interconnected network. The electrical behaviors of the material are crucial for EMI shielding efficiency since it is responsible for interacting with the electromagnetic wave. The total EMI SE is influenced by the mesh size and the amount of mobile charge carriers provided by the filler network in the composites. The EMI SE results of these nanocomposites showed higher values than the minimum value of EMI SE of the shielding required for practical applications, which are usually rated around 20 dB.

Figure 12a shows that the reflection loss (SE_R_) increased with frequency, from 5.56 to 18.96 dB at low percentages of filler (0.8% RGO). Also, SE_R_ increases gradually from 4.52 to 11.65 dB at high percentage of filler (4% RGO). The absorption loss SE_A_ decreased from 6.30 to 3.61 dB at the low percentage of filler (0.8% RGO), and the (SE_A_) decreased gradually from 6.62 to 3.50 dB at a high percentage of filler (4% RGO) as shown in Figure 12b. Figure 12c indicates the proportional relation between the SE _total_ values to the frequency. The inverse proportional effect of RGO loading on SE_R_ and SE_total_ values is shown in Figure 12d, where the RGO increment reflect a reduction of SE_R_ and SE_total._ While the SE_A_ values are increasing to RGO loading increment. The mean EMI SE values of the composites with 0.8%, 1.60%, 2.4%, 3.2%, and 4% mass fractions of RGO are presented in Table 3.

The EMI SE results of the RGO/PLA/PEG nanocomposites in the current work were compared to the previously reported composites prepared with different mixing approaches, different sample thickness, and similar or even lower/higher conductive filler loading, as listed in Table 4. The results of current RGO/PLA/PEG nanocomposites exhibit an efficient EMI SE even when compared to the best values for different polymer/conductive filler nanocomposites. The minimum value of EMI SE of the shielding required for practical application is usually considered to be ~20 dB.

### 3.7. Power Loss and Affective Absorbance A_eff_

While total shielding effectiveness is an important parameter commonly used to quantify the efficiency of a shielding material, it does not provide information on the contributions of each of the shielding mechanisms. To determine the influence of RGO powder loading on the reflection and absorption of the nanocomposites, power balance calculations were performed at various frequencies, which can be expressed by following the generalized equation;
(6)$$\mathrm{Power}\;\mathrm{loss}=1-S_{11}^{2}-S_{21}^{2}$$

The power loss results of the RGO/PLA/PEG nanocomposites displayed in Figure 13 show that the low values of power loss in the case of RGO/PLA/PEG nanocomposites containing high amounts of RGO are due to the low power transmitted into the sample as a result of a very good reflection [64] at the sample’s surface. A better understanding of RGO/PLA/PEG nanocomposites’ potential to absorb electromagnetic radiation can be obtained by evaluating their effective absorbance. The intensity of an EM wave inside the material after primary reflection is based on quantity (1-R), which can be used for adjustment of absorbance (A) to yield effective absorbance from the formula [65]: *A_eff_* (%) = (1 − R − T)/(1 − R) × 100
(7)

*A_eff_* determines what percentage of the power entering the material has been absorbed. Therefore, it is essential to differentiate between the absolute amount of power absorbed by a shielding material and its potential for absorption. Materials that have high reflection can be used for EMI shielding in cases where reflection is of no concern. On the other hand, materials that have high absorption potential (SE_A_) but also reflect a significant fraction of the incident power at their surface may be used as radiation absorbers if their reflection could be reduced via impedance matching at the surface as in structure-engineered shielding, such as multilayer structures or via foaming, etc. [66]. Figure 14 shows the variation of effective absorbance values to the frequency in the RGO/PLA/PEG nanocomposites at different RGO loading. Where the *A_eff_* values of all the samples decrease to the frequency rise, whereas, the higher *A_eff_* values have been observed at the higher RGO loading in the PLA/PEG polymer matrix.

## 4. Conclusions

Reduced graphene oxide (RGO)-reinforced polylactic acid (PLA)/polyethylene glycol (PEG) blended nanocomposites were prepared by the melt blending method. FE-SEM micrographs showed good dispersion of RGO nanoparticles in the PLA/PEG matrix at low concentrations, while at higher loadings, RGO was found to become physically in contact, forming a conductive passage inside the matrix. X-ray diffraction shows that the addition of RGO did not considerably affect the crystallinity of the resulting nanocomposite materials. The improved thermomechanical properties of the composites yielded to a good adhesion between the RGO nanoparticles and the PLA/PEG matrix. The RGO acts as a heat barrier, which enhances the overall thermal stability of the polymer nanocomposites and assists in the formation of ash after thermal decomposition. The dielectric properties of the PLA/PEG matrix were enhanced remarkably with the addition of RGO. The EMI shielding properties of the synthesized composites were tested and the composites showed that the SE _total_ value is higher than the target value of 20 dB. Reflection was found to be the dominant shielding mechanism for RGO/PLA/PEG nanocomposites over the X band. However, the contribution of absorption to SE_total_ increased as the RGO loading was decreased. The effective absorbance of the matrix increased with loading increase. The results showed increased power loss with an increase in the frequency and conversely decreased with an increased percentage of filler. Effective absorbance of the materials increased with loading increase. The materials showed high absorption potential (SE_A_), but also reflected a significant fraction of the incident power at their surface. Thus, the materials may be used as radiation absorbers if their reflection could be reduced via impedance matching at the surface, as in structure-engineered shielding, such as multilayer structures or via foaming.

## Figures and Tables

**Figure 1 polymers-11-00661-f001:**
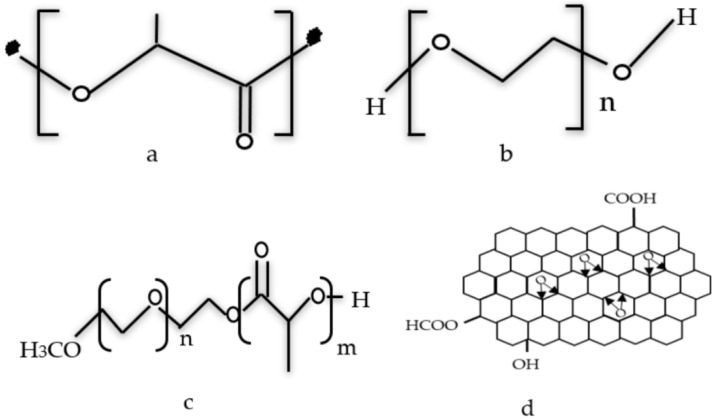
Chemical structures of (**a**) pure PLA, (**b**) pure PEG, (**c**) PLA/PEG mixture, and (**d**) RGO powder.

**Figure 2 polymers-11-00661-f002:**
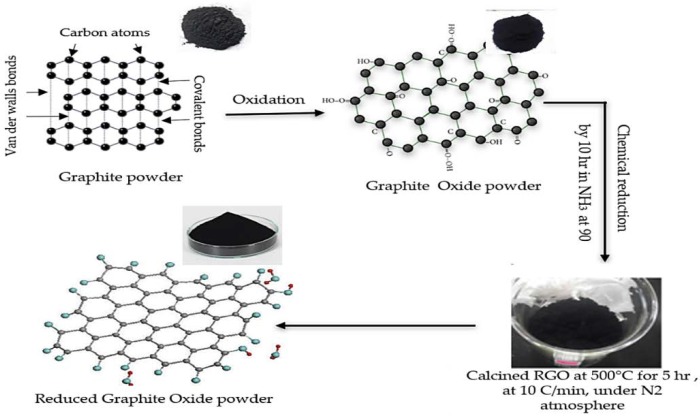
Schematic drawings for the preparation of reduced graphene oxide powder.

**Figure 3 polymers-11-00661-f003:**
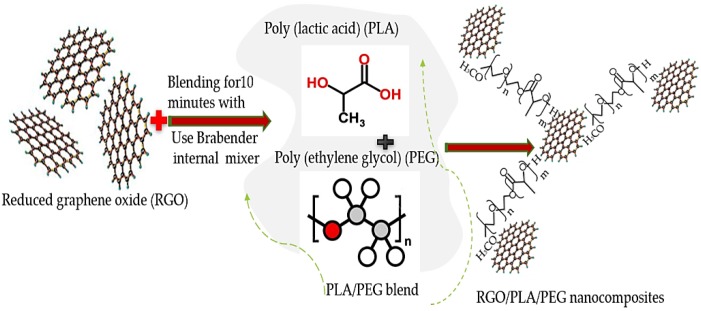
Schematic diagram for the synthesis of RGO/PLA/PEG nanocomposite.

**Figure 4 polymers-11-00661-f004:**
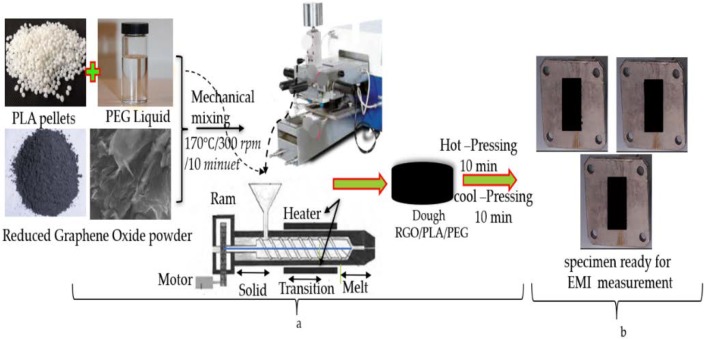
Process for (**a**) the preparation of RGO/PLA/PEG nanocomposites and (**b**) the electromagnetic measurement specimens.

**Figure 5 polymers-11-00661-f005:**
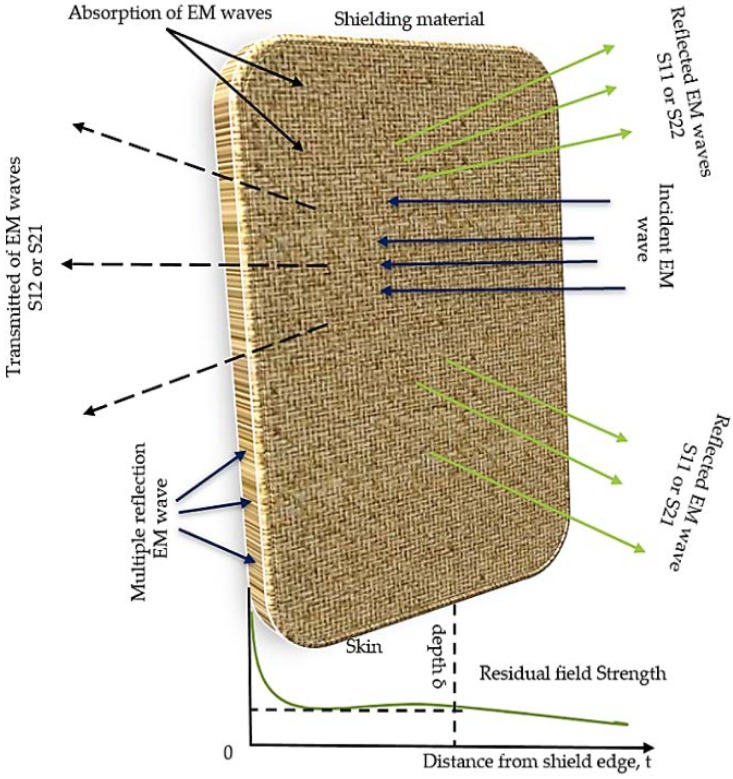
Graphical representation of incident electromagnetic (EM) wave interaction with shield material.

**Figure 6 polymers-11-00661-f006:**
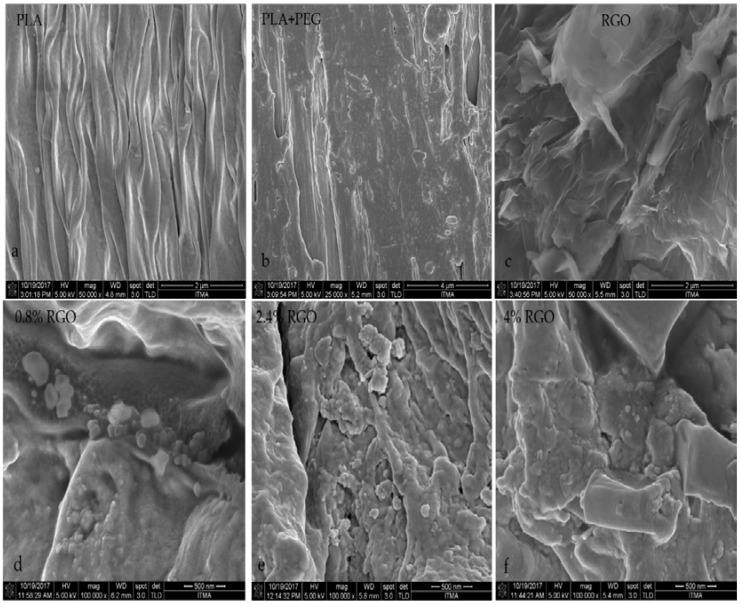
The field emission-scanning electron microscopy (FE-SEM) micrographs of the (**a**) pure PLA, (**b**) PLA/PEG mixture, (**c**) RGO powder with different RGO loadings of (**d**) 0.8%, (**e**) 2.4%, and (**f**) 4% RGO.

**Figure 7 polymers-11-00661-f007:**
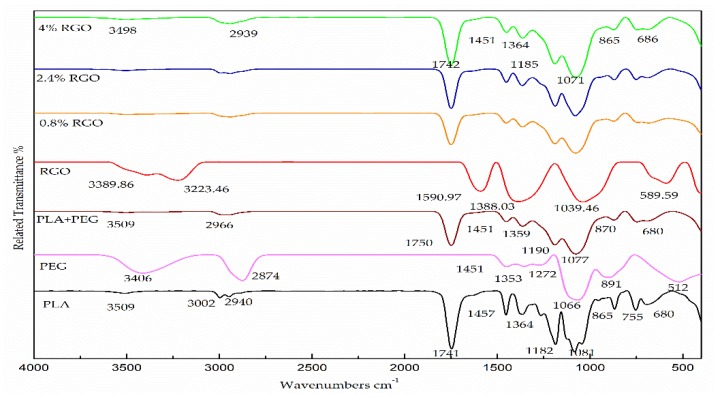
The Fourier transform infrared (FT-IR) bands of the pure PLA, pure PEG, PLA/PEG mixture, and the RGO/PLA/PEG nanocomposites at different percentages of filler.

**Figure 8 polymers-11-00661-f008:**
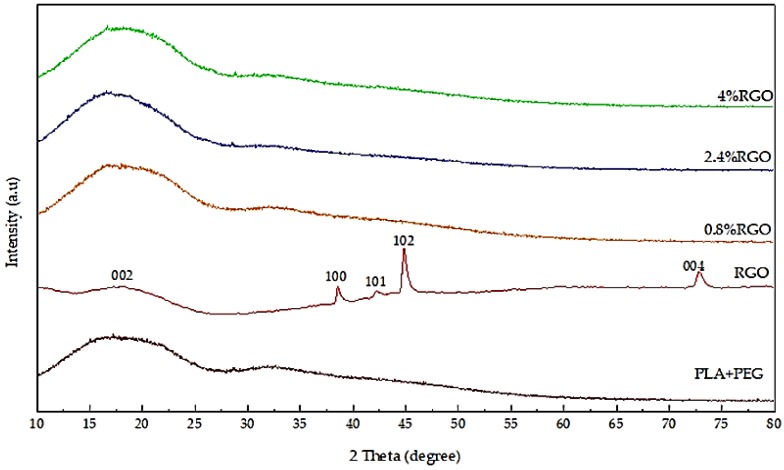
The X-ray diffraction (XRD) patterns of the PLA/PEG mixture, RGO powder, and RGO/PLA/PEG nanocomposites at different percentages of RGO.

**Figure 9 polymers-11-00661-f009:**
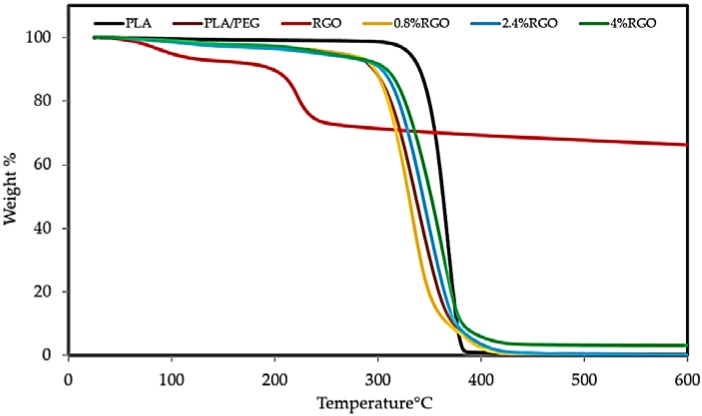
Thermogravimetric analysis (TGA) to study the effect different additions of RGO nanoparticles on thermal stability of the PLA/PEG mixture.

**Figure 10 polymers-11-00661-f010:**
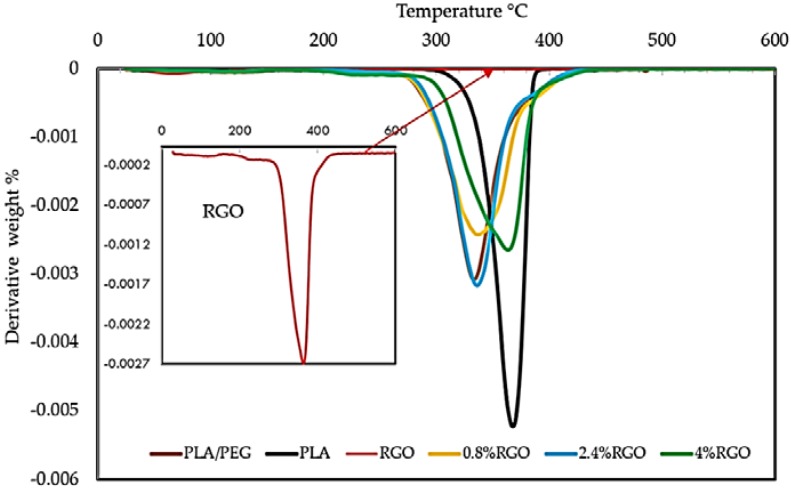
Derivative thermogravimetric (DTG) curves to study the effect of different additions of RGO nanoparticles on the thermal stability of the PLA/PEG mixture.

**Figure 11 polymers-11-00661-f011:**
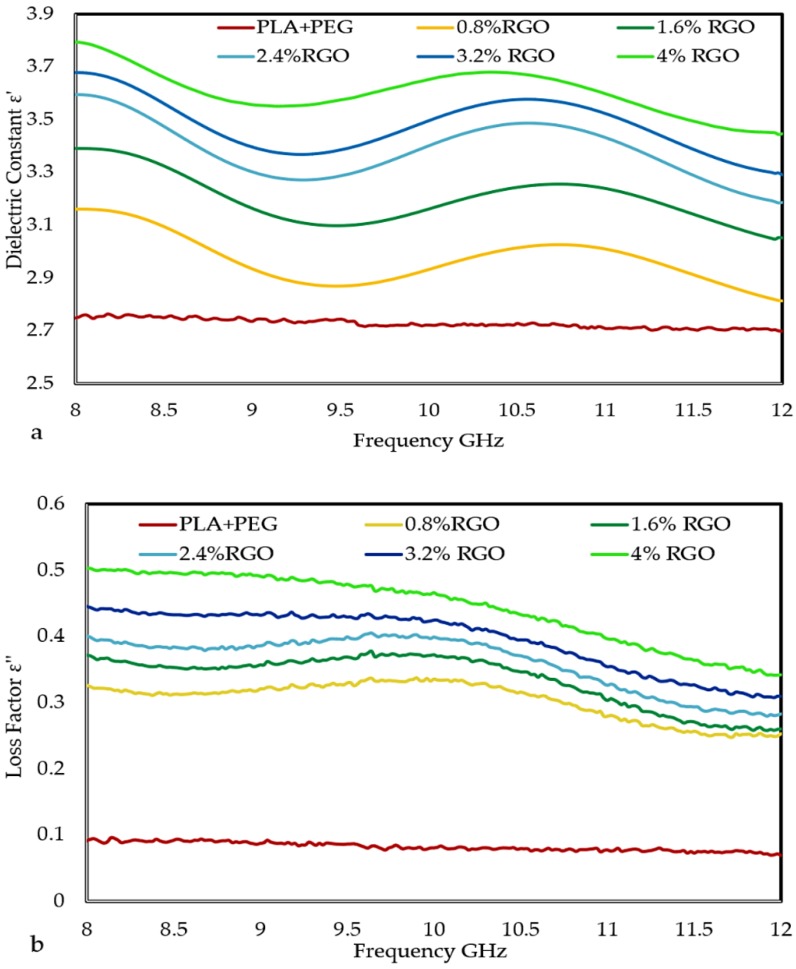
The frequency dependence of (**a**) dielectric constant (ε′), (**b**) loss factor (ε″) at various RGO loadings.

**Figure 12 polymers-11-00661-f012:**
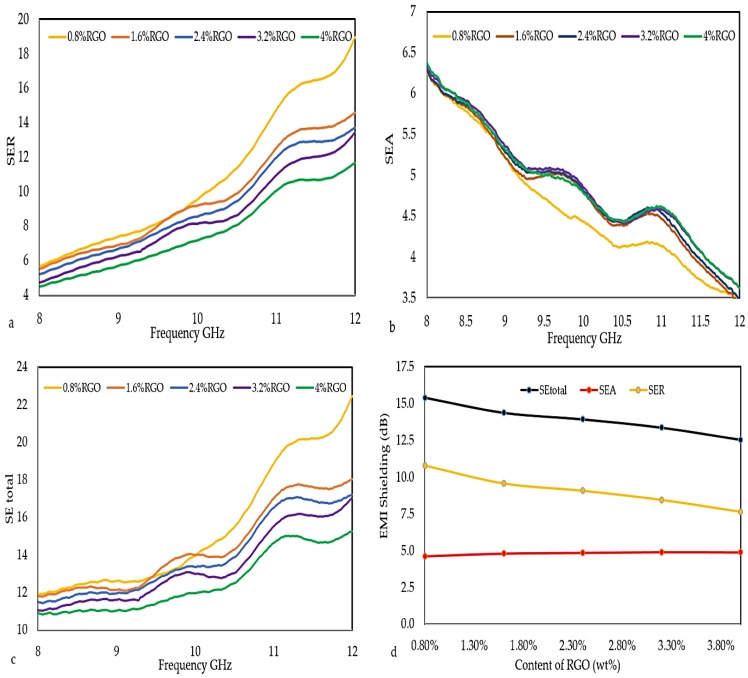
Variation of EMI Shielding Efficiency (EMI SE) against frequency (Hz) for (**a**) SE_R_, (**b**) SE_A_, (**c**), SE_total_, and (**d**) The Shielding Effectiveness (SE) at versus RGO loadings.

**Figure 13 polymers-11-00661-f013:**
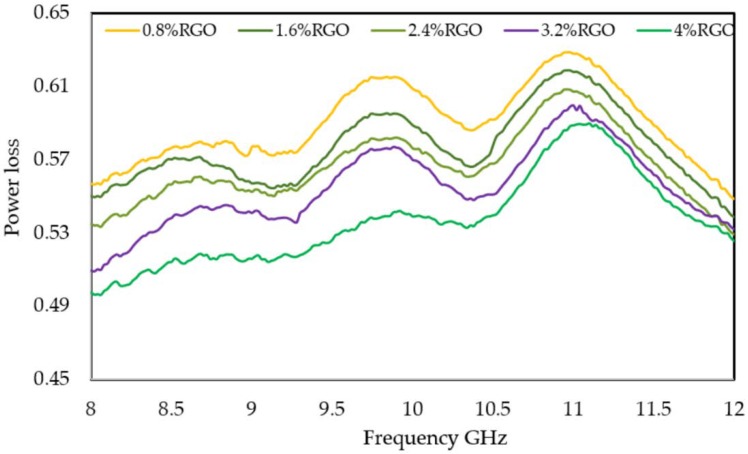
Power loss versus RGO loading over the X band.

**Figure 14 polymers-11-00661-f014:**
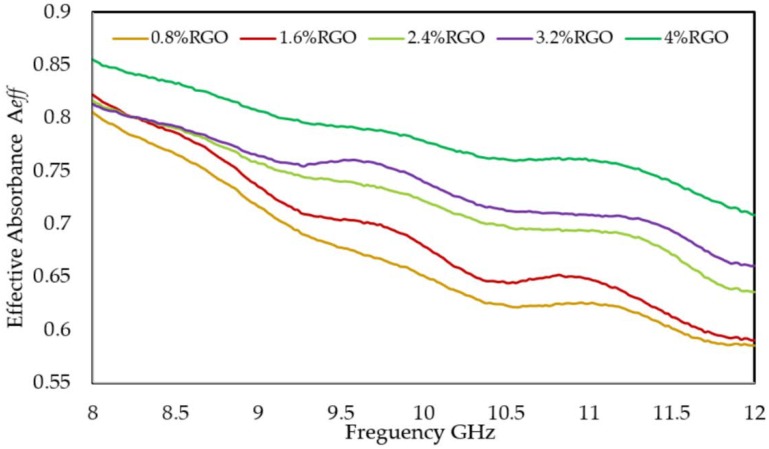
Effective absorption versus RGO loading over the X band.

**Table 1 polymers-11-00661-t001:** Composition of RGO/PLA/PEG nanocomposites.

Sample	Weight of PLA		Weight of PEG		Filler Contents of RGO		Mass (g) RGO/PLA/PEG
RGO/PLA/PEG	g	%	g	%	g	%	25 gm
	22.5	90.00	2.5	10	0	0	
	22.32	89.28	2.48	9.92	0.2	0.8	
	22.14	88.56	2.46	9.84	0.4	1.6	
	21.96	87.84	2.44	9.76	0.6	2.4	
	21.78	87.12	2.42	9.68	0.8	3.2	
	21.6	86.4	2.4	9.60	1	4	

**Table 2 polymers-11-00661-t002:** Summary of the main results of the thermal degradation and the weight loss (%) of neat PLA, neat RGO powder, PLA/PBAT blend (90/10 wt/wt), and RGO/PLA/PEG nanocomposites.

Samples	T_50%_ °C	T_d-max_ °C	Loss Weight (%) T_d-mix_
PLA	365	387.33	97
PLA/PEG	335	372.0	95
RGO	227	241.83	27
0.8% RGO	330	359.12	92
2.4% RGO	340	378.167	90
4% RGO	350	386	87

**Table 3 polymers-11-00661-t003:** The mean EMI SE of RGO/PLA/PEG nanocomposites at different RGO loadings.

Percentage of RGO Filler	SE_total_	SE_A_	SE_R_
0.8	15.41	4.62	10.78
1.6	14.38	4.81	9.57
2.4	13.94	4.85	9.08
3.2	13.37	4.90	8.47
4	12.55	4.89	7.66

**Table 4 polymers-11-00661-t004:** Results of shielding effectiveness of commercial materials based on various polymer composites reported by previous authors in different bands.

Composites	Filler Content	Mixing Procedure	Thickness (mm)	Frequency (GHz)	EMI SE_total_ (dB)	Reference
Epoxy/reduced graphene oxide	15 wt %	Solution	2	8.2–12.4	21	[57]
Polystyrene/reduced graphene oxide	10 wt %	Solution	2.8	8.2–12.4	18	[58]
Epoxy/Single walled carbon nanotube	15 wt%	Solution	2	8.2–12.4	23	[59]
Polystyrene/Vapor grown carbon nanofiber	7 wt %	Solution	1	10	8	[60]
Multi walled carbon nanotube/Cellulose Triacetate	40 wt %	Solution	6	8.2–12.4	30	[61]
Carbon paste electrode/carbon nanofibers	10 wt %	Melt	1	8.2–12.4	22.5	[62]
Reduced graphene oxide/cured epoxy	5% wt %	Solution	6	8.2–12.4	25.748	[35]
Sulfanilic acid azocromotrop/reduced graphene oxide/epoxy	0.5 wt %	Solution	-------	2–8	22.6	[63]
Reduced graphene oxide/poly(lactic acid)/poly(ethylene glycol)	0.8 wt %	Melt	6	8.2–12.4	22.5	Current work

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
