# Peer review of "Chemically Reduced Graphene Oxide-Reinforced Poly(Lactic Acid)/Poly(Ethylene Glycol) Nanocomposites: Preparation, Characterization, and Applications in Electromagnetic Interference Shielding"

_polymers, 2019, doi:10.3390/polym11040661_

Round 1

Reviewer 1 Report

The authors prepared rGO/PLA/PEG nanocomposites by the melt blending method. They revealed the binding mode of the mixture as well as its EMI-related properties with different characterizations. Minor revision is suggested and a few comments in details:

1. It’s suggested to keep all the abbreviations consistent, such as RGO/rGO.

2. It’s better to draw a schematic diagram to help explaining the structure of rGO/PLA/PEG composites. 

3. On page 8 some of the figure numbers are mismatched.

4. Considering this paper is extracted from a dissertation/thesis, it’s suggested to streamline the content. The experimental details section could be more concise. The conclusion from FTIR and XRD is a bit overlapped. 

5. In the TGA, it’s necessary to do component percentage analysis after heating. It seems the ratio of rGO/PLA/PEG will change, then the status of samples has actually been changed too, which means it would be hard to maintain the performance even if you have found an optimized ratio.

6. In section 3.5/3.6/3.7, each component (rGO, PLA, PEG) should be set as control groups to verify these materials have a synergistic effect in the composites.

7. How do rGO/PLA/PEG composites compare with current commercial materials in EMI shielding properties?

Author Response

List of Responses to the Reviewer(s) Comments on the Manuscript (Polymers-454106)

Chemically Reduced Graphene Oxide Reinforced poly (lactic acid)/ Poly (ethylene glycol) Nanocomposites: Preparation, Characterization, and Applications in Electromagnetic Interference Shielding

Thank you for reviewing our submission and your very valuable comments. We carefully responded to all the points and have modified the manuscript accordingly.

The following are the responses to the comments and Suggestions of Reviewers: -

(A)           The first reviewer in this manuscript

(x) English language and style are fine/minor spell check required

In general, the manuscript has been checked it linguistically, content, references, and formatting to achieve the publishing requirements of your journal.      

(x)Is the research design appropriate?

 The research design has been modified appropriately to achieve the publishing requirements of your journal.

 1- It’s suggested to keep all the abbreviations consistent, such as RGO/ rGO.

All the abbreviations of RGO have been made uniform and consistent,

It’s better to draw a schematic diagram to help explaining the structure of RGO/PLA/PEG nanocomposites. 

The schematic diagram to explaining the structure of RGO/ PLA/ PEG nanocomposites was added in section ''2.3.  Preparation of RGO/ PLA/PEG 

nanocomposites'' Line 144

Reviewer 2 Report

The authors report a facile method to produce rGO/poly 2 (lactic acid)/ Poly (ethylene glycol) Nanocomposites with enhanced dielectric properties and electromagnetic interference shielding effectiveness. The reinforcement mechanism is systematically investigated via FT-IR, FE-SEM, XRD, and TGA. The manuscript can be accepted for publication once the following comments are properly addressed.

1) It is believed that the GO morphologies such as size and geometry can play an important role in determining the final dielectric and EMI shielding properties. Unfortunately, the related information is missing.

2) The EMI SE value of around 20 dB is not outstanding. Please make a comparison to highlight the new contribution of this work.

Author Response

List of Responses to the Reviewer(s) Comments on the Manuscript (Polymers-454106)

Chemically Reduced Graphene Oxide Reinforced poly (lactic acid)/ Poly (ethylene glycol) Nanocomposites: Preparation, Characterization, and Applications in Electromagnetic Interference Shielding

Thank you for reviewing our submission and your very valuable comments. We carefully responded to all the points and have modified the manuscript accordingly.

The following are the responses to the comments and Suggestions of Reviewers: -

(A) The second reviewer in this manuscript

(XModerate English changes required 

In general, the manuscript has been checked linguistically including the content, references, and formatting to achieve the publishing requirements of your journa

l.     

(X) Are the methods adequately described?

The method description of nanocomposite’s preparation in the manuscript has been checked and modified.

(X) Are the results clearly presented?

The results section at the manuscript has been modified. 

1)   It is believed that the GO morphologies such as size and geometry can play an important role in determining the final dielectric and EMI shielding properties. Unfortunately, the related information is missing.

Extra references have been added in the introduction Section that explains the effect of the particle size on the dielectric and EIM (SE) properties, Line (102-113). 

‘’The electromagnetic properties and EMI shielding effectiveness performances of 

nanocomposites depend on the reflection from the material's surface, absorption of the EM energy, and propagation paths of the EM wave, which are determined by nature, shape, size, and microstructure of the fillers [wen 2014].  There are some studies that have investigated and contributed to the knowledge of, the effect of the particle size of the graphene oxide, on the electromagnetic properties, where the properties can be tuned by controlling size of the graphene particles. This phenomenon may be interpreted as follows: the interfacial polarization is dominant in the heterogeneous structures, which contributes to the various electromagnetic properties [Zhang 2010]. Also, some defects existing graphene oxide act as the dipolar polarization centers, leading to various dielectric relaxations in different frequency ranges.  With decreasing the particle size of the nanoparticle, larger interface area and more defects exist in the nanoparticles, which enhance the electromagnetic properties of the nanoparticles. [ahmad et all 33].”

2) The EMI SE value of around 20 dB is not outstanding. Please make a comparison to highlight the new contribution of this work.

The current EMI shielding effectiveness results have been compared with the results of the previous studies and have been added to the manuscript at section’’3.6. EMI shielding effectiveness’’ Line 437- 445

‘’The EMI SE total results of the RGO/PLA/PEG nanocomposites in current work was compared to the previously reported composites at different mixing approaches, different sample thickness and similar or even lower/higher conductive filler loading, as listed in Table 4. The results of current RGO/PLA/PEG nanocomposites exhibit an efficient EMI SE even when compared to the best values for different polymer/conductive filler nanocomposites. The minimum value of EMI SE of the shielding required for practical application is usually considered to be ~20 dB.’’ 

Table 4. Results of shielding effectiveness of commercial materials based on various polymer composites reported by previous authors in different bands.

Composites

Filler content

Mixing procedure

Thickness

(mm)

Frequency (GHz)

EMI SE total (dB)

Reference

Epoxy/ reduced   graphene oxide

15 wt%

Solution

2

8.2-12.4

21

57

Polystyrene/ reduced   graphene oxide

10 wt%

Solution

2.8

8.2-12.4

18

58

Epoxy/ Single   walled carbon nanotube

15 wt%

Solution

2

8.2-12.4

23

59

Polystyrene/ Vapor   grown carbon nanofiber

7 wt%

Solution

1

10

8

60

Multi walled carbon   nanotube/Cellulose Triacetate

40 t%

Solution

6

8.2-12.4

30

61

Carbon paste   electrode/ carbon nanofibers

10 wt%

Melt

1

8.2-12.4

22.5

62

Reduced graphene   oxide/ cured epoxy

5% wt%

Solution

6

8.2-12.4

25.748

35

Sulfanilic acid

azocromotrop/reduced   graphene oxide /epoxy

0.5 wt%

Solution

­­­­­­­­­­­­­­-------

2–8

22.6

63

Reduced graphene   oxide / poly (lactic acid) /poly (ethylene glycol)

0.8 wt %

Melt

6

8.2-12.4

22.5

Current work

If you need further clarifications or modification, you are more than welcome to contact me at “ahmad_al67@yahoo.com” or ahmadfahad@upm.edu.my.

• All the comments requested by the reviewers were acted upon.

Thank You.

Ahmad Fahad Ahmad (Ph.D)

Physics Department, Faculty of Science, Universiti Putra Malaysia

 ahmad_al67@yahoo.com

ahmadfahad@upm.edu.my

+60 173370907

Round 2

Reviewer 2 Report

It can be accepted for publication.